# Impact of the Substitution Pattern at the Basic Center and Geometry of the Amine Fragment on 5-HT_6_ and D_3_R Affinity in the 1*H*-Pyrrolo[3,2-*c*]quinoline Series

**DOI:** 10.3390/molecules28031096

**Published:** 2023-01-21

**Authors:** Katarzyna Grychowska, Wojciech Pietruś, Ludmiła Kulawik, Ophélie Bento, Grzegorz Satała, Xavier Bantreil, Frédéric Lamaty, Andrzej J. Bojarski, Joanna Gołębiowska, Agnieszka Nikiforuk, Philippe Marin, Séverine Chaumont-Dubel, Rafał Kurczab, Paweł Zajdel

**Affiliations:** 1Faculty of Pharmacy, Jagiellonian University Medical College, 9 Medyczna Str., 30-688 Kraków, Poland; 2Maj Institute of Pharmacology, Polish Academy of Sciences, 12 Smętna Str., 31-324 Kraków, Poland; 3Institut de Génomique Fonctionelle, Université de Montpellier, CNRS INSERM, 34094 Montpellier, France; 4IBMM, Université de Montpellier, CNRS, ENSCM, 34094 Montpellier, France; 5Institut Universitaire de France (IUF), 75005 Paris, France

**Keywords:** 5-HT_6_R antagonists, D_3_R ligands, dual-acting compounds, molecular dynamics, salt bridge formation

## Abstract

Salt bridge (SB, double-charge-assisted hydrogen bonds) formation is one of the strongest molecular non-covalent interactions in biological systems, including ligand–receptor complexes. In the case of G-protein-coupled receptors, such an interaction is formed by the conserved aspartic acid (D3.32) residue and the basic moiety of the aminergic ligand. This study aims to determine the influence of the substitution pattern at the basic nitrogen atom and the geometry of the amine moiety at position 4 of 1*H*-pyrrolo[3,2-*c*]quinoline on the quality of the salt bridge formed in the 5-HT_6_ receptor and D_3_ receptor. To reach this goal, we synthetized and biologically evaluated a new series of 1*H*-pyrrolo[3,2-*c*]quinoline derivatives modified with various amines. The selected compounds displayed a significantly higher 5-HT_6_R affinity and more potent 5-HT_6_R antagonist properties when compared with the previously identified compound **PZ-1643**, a dual-acting 5-HT_6_R/D_3_R antagonist; nevertheless, the proposed modifications did not improve the activity at D_3_R. As demonstrated by the in silico experiments, including molecular dynamics simulations, the applied structural modifications were highly beneficial for the formation and quality of the SB formation at the 5-HT_6_R binding site; however, they are unfavorable for such interactions at D_3_R.

## 1. Introduction

Serotonin type 6 receptor (5-HT_6_R) belongs to the family of G-protein-coupled receptors (GPCRs), which has emerged as a promising target for the treatment of cognitive decline associated with neurodegenerative (e.g., Alzheimer’s disease and Parkinson’s disease) and psychiatric disorders (e.g., depression and schizophrenia).

Apart from coupling to the Gs protein, 5-HT_6_R participates in other signaling pathways [1], including the mechanistic target of rapamycin (mTOR) [2], which accounts for the impact of the receptor in some cognition paradigms in rodents and cyclin-dependent kinase 5 (Cdk5) [3], which is involved in the neurogenesis process. One of the characteristic features of this receptor is its high level of constitutive activity, defined as the spontaneous activity of the receptor in the absence of an agonist [4]. In the hippocampus and the frontal cortex, 5-HT_6_Rs are localized on the neuronal dendrites and primary neuronal cilia of glutamatergic, GABA-ergic, and cholinergic neurons [5,6]. The ciliary location is of particular interest, as these sensory organelles are implicated in the neurodevelopmental process. The pharmacological blockade of 5-HT_6_R enhances cholinergic and glutamatergic neurotransmission, indicating that this mechanism is engaged in the improvement of cognitive functions displayed by 5-HT_6_R antagonists in animal models [7,8].

Recently, the development of dual-acting agents, which not only could relieve cognitive decline but may also produce antidepressant and anxiolytic effects [9] and antipsychotic [10] and neuroprotective properties [11], has gained considerable attention. A number of compounds combine antagonism at the 5-HT_6_R with serotonin type 2A receptor (5-HT_2A_R) antagonism [12,13], serotonin type 3 receptor (5-HT_3_R) antagonism [10], serotonin type 4 receptor (5-HT_4_R) agonism [14,15], GABA-A agonism [16] acetylcholinesterase inhibition [17], or monoaminoxidase type B (MAO-B) inhibition [18]

Simultaneous blockade of the dopamine D_3_ receptor (D_3_R) is one of the promising strategies in the elaboration of 5-HT_6_R antagonism-based dual-acting compounds for improved treatment of Alzheimer’s disease and other neurodegenerative disorders [19,20]. D_3_R is a GPCR localized in the limbic areas of the brain [21]. In addition to coupling to the G_i_/_0_ protein, it additionally engages the Cdk5 [22] and mTOR [23] pathways, leading to the enhancement of acetylcholine and glutamate signaling [19,24].

A quest for dual-acting 5-HT_6_/D_3_Rs antagonists has been initiated by the identification of compound SB737050 (Figure 1); however, it was burdened with a relatively high affinity for D2Rs [25]. Subsequently, a 2,3,4,7-tetrahydro-1*H*-pyrrolo[2,3-*h*]isoquinoline derivative **I** displaying a more balanced profile for 5-HT_6_ and D_3_Rs was described [19].

Recently disclosed compound **PZ-1643**, assigned as derivative 19 in [26], a dual-acting 5-HT_6_/D_3_R antagonist, was designed as a merged ligand in which the selective 5-HT_6_R neutral antagonist CPPQ ((S)-1-((3-chlorophenyl)sulfonyl)-*N*-(pyrrolidin-3-yl)-1*H*-pyrrolo[3,2-*c*]quinolin-4-amine, disclosed as compound 14 in [27], was combined with an alkyl chain, representing a characteristic structural feature of D_3_R antagonists.

To further investigate the impact of the kind of the substituent at the basic nitrogen atom and the geometry of the amine fragment on 5-HT_6_R and D_3_R affinity, we designed a small series of compound **PZ-1643** analogs. Structural modifications comprised the introduction of various alkyl-derived chains on the basic nitrogen atom and replacement of (*S*)-3-amino-1-Boc-pyrrolidine with enantiomers of 2-(aminomethyl)pyrrolidines and 3-(aminomethyl)pyrrolidines (Figure 2). The affinity for both targets was assessed in the binding experiments at 5-HT_6_R and D_3_R and was confirmed by molecular dynamics (MD) evaluation, which determined the quality of the salt bridge (SB) formed with D3.32 of 5-HT_6_R and D_3_R.

## 2. Results and Discussion

### 2.1. Chemistry

The key 1*H*-pyrrolo[3,2-*c*]quinoline synthon **5** was obtained in a multistep synthesis route, following the previously reported protocol (Figure 1) [28].

Heating of compound **5** with the excess of respective amine under microwave-assisted conditions yielded Boc-protected amine derivates, **6a**–**6e**, which were further coupled with 3-chlorobenzenesulfonyl chloride in the presence of a phosphazene base yielding sulfonyl derivatives **7a**–**7e** (Figure 2). Treatment with 1M HCl in methanol afforded secondary amines, for which reductive amination was carried out with respective aldehydes.

### 2.2. Determination of Affinity of Compounds for 5-HT_6_R and D_3_R and Assessment of the Impact of the Selected Compounds on 5-HT_6_R-Dependent Gs Signaling

The biological evaluation was initiated by assessing the affinity of the compounds for 5-HT_6_R in the [^3^H]-LSD radioligand binding assay. Experiments were performed in a stable HEK293 cell line expressing the human 5-HT_6_R [10]. The selected compounds, which showed the highest affinity for the serotoninergic target, were further tested for their affinity for D_3_R in the screening procedure using [^3^H]-methylspiperone as the radioligand. Experiments were performed in Chinese hamster ovary (CHO) cells with the stable expression of human D_3_R (Eurofins, Celle-L’Evescault, France) [29].

Antagonist properties of the most active compounds at 5-HT_6_R were evaluated in cAMP cellular assays, and their impact on cAMP production induced by 5-CT was studied [30]. The experiments were performed in 1321N1 cells expressing the human serotonin 5-HT_6_R. Finally, the impact of the selected derivatives on agonist-independent 5-HT_6_R-operated Gs signaling was tested in NG108-15 cells transiently expressing 5-HT_6_Rs, a cellular model in which 5-HT_6_R exhibits a high level of constitutive activity [5].

### 2.3. Structure–Activity Relationship Analysis

Designing dual-acting 5-HT_6_/D_3_R ligands in a group of 1*H*-pyrrolo[3,2-*c*]quinolines revealed that the introduction of an isobutyl chain on the nitrogen atom of pyrrolidine present in compound **PZ-1643** maintained the high affinity for 5-HT_6_R and was beneficial for the affinity for D_3_R [26]. The analysis of molecular dynamics (MD) of the pairs of R and S enantiomers indicated that the S counterpart showed beneficial parameters for the distance and angle of the SB in 5-HT_6_R.

In the present study, initial efforts comprised the replacement of the isobutyl chain of the lead compound **PZ-1643** (Figure 2) [26], with more polar substituents (Table 1). Encouragingly, the introduction of 2-hydroxyprop-1-yl enantiomers significantly increased the affinity for 5-HT_6_R (**8**, **9** vs. **PZ-1643**). Additionally, a preference for the R enantiomer was observed. The introduction of the 2-hydroxyethyl moiety was unfavorable for the interaction with 5-HT_6_R and decreased the affinity by threefold (**10** vs. **PZ-1643**). Replacement of the hydroxyl group of **10** with a more hydrophobic trifluoromethyl substituent further decreased the 5-HT_6_R affinity (**12** vs. **10**). However, the introduction of 3-methoxyprop-1-yl was well tolerated (**11**). Being the most active compound from the evaluated series, **9** displayed a moderate affinity for D_3_R in the screening procedure.

Further studies focusing on the modifications of the geometry of the amine fragment (Table 2) revealed that R enantiomers were preferred for binding to 5-HT_6_R (**13** vs. **14, 15** vs. **16**, **17** vs. **18**, **19** vs. **20**).

Our recent studies revealed that the introduction of an alkyl substituent on the basic nitrogen atom could be beneficial for dual 5-HT_6_R/D_3_R activity. In the evaluated series the ethyl chain on the basic center of (R)-2-(aminomethyl)pyrrolidinyl maintained the affinity for 5-HT_6_R when compared with **PZ-1643 (13** vs. **PZ-1643)**. The same modification applied in the 3-(aminomethyl)pyrrolidinyl derivatives decreased the affinity for 5-HT_6_R (**15**, **16** vs. **PZ-1643**) and indicated the preference of the (R)-2-(aminomethyl)-congener in the interaction with a serotoninergic target. Therefore, 2-(aminomethyl)pyrrolidinyl was selected as the amine fragment for further diversification and evaluation for 5-HT_6_R and D_3_R.

Further structural diversification at the basic nitrogen atom, which involved the replacement of ethyl with a more bulky isobutyl substituent in the R series of 2-(aminomethyl)pyrrolidine (**17**), maintained the affinity for 5-HT_6_R compared with compound **PZ-1643**. Subsequent modification, which comprised the introduction of 2-hydroxyethyl and 3-methoxyprop-1-yl, was unfavorable for the interaction with 5-HT_6_R (**19**, **20**, **21** vs. **PZ-1643**).

Most active compounds of the evaluated series were functionalized with (R)-2-(aminomethyl)pyrrolidine at position 4 of the 1*H*-pyrrolo[3,2-*c*]quinoline core. Among them, derivatives containing an amine fragment substituted with ethyl (**13**) or isobutyl chains (**17**) displayed moderate affinity for D_3_R. Therefore, the applied structural modifications were favorable for the 5-HT_6_R affinity compared with **PZ-1643**; however, they did not improve the affinity for D_3_R.

Taking into account the high affinity of **13** and **17** for 5-HT_6_R and the most potent affinity for D_3_Rs among the evaluated compounds, these derivatives were further tested for their antagonist properties at 5-HT_6_R in cellular assays performed in recombinant 1321N1 cells expressing the human serotonin 5-HT_6_R. The results of these assays were in line with those obtained from the binding experiments, since the evaluated compounds displayed nanomolar antagonist properties at 5-HT_6_R (**13**, *K*_b_ = 1.2 nM; **17**, *K*_b_ = 3.8 nM). Further evaluation of the impact of the selected derivatives on 5-HT_6_R-operated Gs signaling revealed their neutral antagonist properties in this pathway (Figure 3) [10].

### 2.4. In Silico Studies

To further investigate the influence of the kind of the substituent at the basic nitrogen atom and geometry of the amine fragment on the SB parameters (i.e., distance and angle) and receptor affinity, the molecular mechanism of action of selected structurally diverse compounds **9**, **12, 13,** and **PZ-1643** was evaluated by flexible molecular docking (IFD procedure) to 5-HT_6_R (PDB ID: 7XTB) and D_3_R (PDB ID: 3PBL). The binding modes were coherent with our previously reported results [26] but did not show clear explanation of the structure–activity relationships (Appendix A). Therefore, a series of MD simulations was performed to determine the features responsible for changes in the potency.

A closer inspection of the MD trajectories showed that various alkyl chains on the nitrogen atom penetrated the narrow hydrophobic subpocket formed by transmembrane (TM) helixes 2, 3, and 7 in both receptors. As we proposed in our previous study, the higher binding activity of different derivatives as well as enantiomers originated from the quality of the SB formed with D3.32 [26,31,32]. Thus, the geometric parameters for the interaction with D3.32 observed during molecular dynamics simulations were analyzed (Table 3).

Regarding the interaction with 5-HT_6_R, compounds **PZ-1643**, **9**, and **13** displayed the most favorable mean geometric parameters of the SB with D3.32 (i.e., both the distance and the angle of the SB lie in the favorable area of the interaction energy) [33]. In addition, **9** showed a hydrogen bond of the -OH group with T3.29. Molecular dynamics results further showed that the SB is a highly stable interaction, with a frequency of occurrence of more than 89% in complexes (Table 3). However, compound **12**, bearing a 3,3,3-trifluoropropyl chain, showed an acceptable geometry of an SB, but in this case, a part of the salt bridge geometry might be distorted by the unfavorable polar interactions formed by the –CF_3_ group and the side chain of Y7.43 and T3.29 (the most distorted side chain among all of Y7.43; Figure 4A). In addition, significant stabilization of all derivatives by the hydrogen bond between S5.44 and the oxygen of sulfonamide groups and the halogen bond formed between the chlorine substituent and the carbonyl oxygen of S4.57 were noted. These interactions are not depicted in Figure 4, since their contribution to the MD trajectory, depending on the derivative, was between 20–40%.

In the case of D_3_R, only **PZ-1643** displayed the favorable geometric parameters of the SB. As revealed by MD simulations, the introduction of a 2-hydroxypropyl or 3,3,3-trifluoropropyl chain on the basic nitrogen atom led to the higher distortion during molecular dynamics, and thus less stable complexes (frequencies of the SB occurrence were substantially lower than for **PZ-1643**; Table 3). The conformations of the side chains of the respective amino acids in the D_3_R binding side were more distorted than in 5-HT_6_R.

## 3. Experimental Methods

### 3.1. Chemistry

#### General Method

The synthesis was conducted at room temperature, unless indicated otherwise. Organic solvents (from Chempur, Piekary Śląskie, Poland) were of reagent grade and were used without purification. All reagents (Sigma-Aldrich (Saint Louis, MI, USA), Fluorochem (Glossop, UK) and TCI (Zwijndrecht, Belgium) were of the highest purity. Column chromatography was performed on silica gel Merck 60 (70–230 mesh ASTM, Darmstadt, Germany).

UPLC and MS were carried out on a system consisting of a Waters Acquity UPLC coupled((Waters Corporation, Milford, MA, USA) to a Waters TQD mass spectrometer. All the analyses were carried out using an Acquity UPLC BEH C18 100 × 2.1 mm^2^ column at 40 °C. A flow rate of 0.3 mL/min and a gradient of (0−100) % B over 10 min was used: eluent A, water/0.1% HCOOH and eluent B, acetonitrile/0.1% HCOOH. Retention times, t_R_, were given in minutes. The UPLC/MS purity of all the test compounds and key intermediates was determined to be >95%.

The ^1^H NMR and ^13^C NMR spectra were recorded using JEOL JNM-ECZR 500 RS1 (ECZR version) at 500 and 126 MHz, respectively, as well as Bruker Advance III HD at 400 and 100 MHz, respectively. Chemical shifts are reported in parts per million using deuterated solvent for calibration (CD_3_OD). The J values are given in Hertz (Hz).

Compounds **1**–**5**, **6a**, and **7** were obtained according to the previously reported procedure and the analytical data are in accordance with the literature [26,28]. Compounds **13**–**21** were converted to hydrochloride salts.


*(S)-1-((S)-3-((1-((3-chlorophenyl)sulfonyl)-1H-pyrrolo[3,2-c]quinolin-4-yl)amino)pyrrolidin-1-yl)propan-2-ol 8*


Pale oil, 53% yield, *t*_R_ = 3.84, C_24_H_25_ClN_4_O_3_S, MW 485.00, ^1^H NMR (400 MHz, CD_3_OD) δ (ppm) 1.17 (d, *J* = 6.1, 3H), 1.81–1.96 (m, 1H), 2.35–2.50 (m, 2H), 2.52–2.58 (m, 2H), 2.75–2.88 (m, 1H), 2.90–3.09 (m, 2H), 3.26–3.37 (m, 2H), 3.86–3.99 (m, 1H), 4.77–4.85 (m, 1H), 7.12–7.21 (m, 2H), 7.36–7.47 (m, 2H), 7.51–7.59 (m, 1H), 7.60–7.66 (m, 1H), 7.67–7.73 (m, 1H), 7.75–7.79 (s, 1H), 7.86–7.97 (d, *J* = 3.4 Hz, 1H), 8.71 (d, *J* = 8.3 Hz, 1H); ^13^C NMR (100 MHz, CD_3_OD) δ (ppm) 20.50, 31.44, 49.63, 53.61, 60.66, 63.50, 65.15, 105.92, 114.34, 116.12, 121.49, 122.77, 125.00, 126.5, 127.5, 128.23, 131.01, 134.26, 135.18, 139.60, 146.47, 151.22. Monoisotopic mass: 484.13, [M + H]^+^ = 485.1.


*(R)-1-((S)-3-((1-((3-chlorophenyl)sulfonyl)-1H-pyrrolo[3,2-c]quinolin-4-yl)amino)pyrrolidin-1-yl)propan-2-ol 9*


Pale oil, 57% yield, *t*_R_ = 3.85, C_24_H_25_ClN_4_O_3_S, MW 485.00, ^1^H NMR (400 MHz, CD_3_OD) δ (ppm) 1.18 (d, *J* = 6.26 Hz, 3H), 1.81–1.94 (m, 1H), 2.36–2.48 (m, 2H), 2.48–2.64 (m, 2H), 2.79 (dd, *J* = 9.98, 3.72 Hz, 1H), 2.92–3.09 (m, 2H), 3.33 (dt, *J* = 3.28, 1.59 Hz, 1H), 3.88–3.99 (m, 1 H), 4.80–4.85 (m, 1H), 7.14–7.20 (m, 2H), 7.35–7.44 (m, 2H), 7.54 (d, *J* = 8.02 Hz, 1H), 7.60–7.65 (m, 1H), 7.68–7.73 (m, 1H), 7.75–7.80 (m, 1H), 7.92 (d, *J* = 3.72 Hz, 1H), 8.68 –8.76 (m, 1H), 8.75 (d, *J* = 8.3 Hz, 1H); ^13^C NMR (100 MHz, CD_3_OD) δ (ppm) 20.56, 31.51, 49.56, 52.89, 61.25, 63.46, 65.13, 105.96, 114.34, 116.15, 121.47, 122.77, 125.00, 126.48, 127.54, 128.21, 131.00, 134.25, 135.17, 135.28, 139.58, 146.50, 151.25. Monoisotopic mass: 484.13, [M + H]^+^ = 485.1.


*(S)-2-(3-((1-((3-chlorophenyl)sulfonyl)-1H-pyrrolo[3,2-c]quinolin-4-yl)amino)pyrrolidin-1-yl)ethan-1-ol 10*


Pale oil, 45% yield, *t*_R_ = 3.86 min, C_23_H_23_ClN_4_O_3_S, MW 470.97, ^1^H NMR (400 MHz, CD_3_OD) δ (ppm) 2.23–2.38 (m, 1H), 2.52–2.62 (m, 4H), 3.27–3.37 (m, 2H), 3.72–3.89 (m, 4H), 7.19 (t, *J* = 7.73 Hz, 1H), 7.22–7.28 (m, 1H), 7.37 (q, *J* = 8.15 Hz, 2H), 7.51 (dd, *J* = 8.22, 0.98 Hz, 1H), 7.60 (d, *J* = 8.02 Hz, 1H), 7.70 (s, 1H), 7.81–7.91 (m, 2H), 8.64 (d, *J* = 8.41 Hz, 1H); ^13^C NMR (100 MHz, CD_3_OD) 29.06, 39.05, 49.95, 53.38, 56.33, 58.53, 106.28, 114.01, 123.05, 125.25, 126.56, 128.37, 129.06, 131.20, 134.61, 134.61, 135.36, 139.29, 149.81. Monoisotopic mass: 470.12, [M + H]^+^ = 471.1.


*(S)-1-((3-chlorophenyl)sulfonyl)-N-(1-(3-methoxypropyl)pyrrolidin-3-yl)-1H-pyrrolo[3,2-c]quinolin-4-amine 11*


Pale oil, 30% yield, *t*_R_ = 3.95, C_25_H_27_ClN_4_O_3_S, MW 499.03, ^1^H NMR (400 MHz, CD_3_OD) δ (ppm) 1.76–1.99 (m, 2H), 2.15–2.30 (m, 1H), 2.41–2.58 (m, 1H), 3.03 (s, 2H), 3.11–3.19 (m, 2H), 3.21–3.24 (m, 3H), 3.24–3.29 (m, 1H), 3.41–3.57 (m, 2H), 3.63–3.80 (m, 1H), 4.78–4.83 (m, 1H), 7.02–7.12 (m, 1 H), 7.13–7.21 (m, 1H), 7.36 (dt, *J* = 10.86, 8.07 Hz, 2H), 7.46–7.54 (m, 1H), 7.54–7.65 (m, 2 H), 7.66–7.70 (m, 1 H), 7.83–7.89 (m, 1H), 8.62–8.72 (m, 1H); ^13^C NMR (100 MHz, CD_3_OD) 26.13, 29.22, 49.88, 52.89, 53.64, 57.61, 59.11, 69.62, 105.86, 110.01, 114.48, 116.08, 122.25, 122.96, 125.13, 126.46, 127.86, 128.63, 131.08, 134.39, 135.24, 139.48, 150.52. Monoisotopic mass 498.15, [M + H]^+^ = 499.1.


*(S)-1-((3-chlorophenyl)sulfonyl)-N-(1-(3,3,3-trifluoropropyl)pyrrolidin-3-yl)-1H-pyrrolo[3,2-c]quinolin-4-amine 12*


Pale oil, 30% yield, *t*_R_ = 4.05, C_24_H_22_ClF_3_N_4_O_2_S, MW 522.97, ^1^H NMR (400 MHz, CD_3_OD) δ (ppm) 2.10 (dd, *J* = 12.89, 5.15 Hz, 1 H), 2.45–2.55 (m, 1H), 2.54–2.54 (m, 1H), 2.56–2.65 (m, 2H), 2.90–2.99 (m, 1H), 3.06–3.13 (m, 1H), 3.15–3.22 (m, 2H), 3.37 (ddd, *J* = 10.02, 8.45, 5.30 Hz, 1H), 4.76–4.83 (m, 1H), 7.15–7.22 (m, 2H), 7.35–7.43 (m, 2H), 7.51 (ddd, *J* = 8.16, 2.00, 1.00 Hz, 1H), 7.58–7.62 (m, 1H), 7.64 (dd, *J* = 8.31, 0.86 Hz, 1H), 7.72 (t, *J* = 1.86 Hz, 1H), 7.89 (d, *J* = 3.72 Hz, 1H), 8.68 (dd, *J* = 8.59, 1.15 Hz, 1H); ^13^C NMR (100 MHz, CD_3_OD) 21.64, 30.28, 50.03, 53.00, 59.36, 106.17, 114.24, 116.03, 122.34, 122.98, 125.16, 126.47, 128.66, 131.22, 134.49, 135.27, 139.37, 144.72, 150.53, 175.74, 177.46. Monoisotopic mass 522.11, [M + H]^+^ = 523.1.


*(R)-1-((3-chlorophenyl)sulfonyl)-N-((1-ethylpyrrolidin-2-yl)methyl)-1H-pyrrolo[3,2-c]quinolin-4-amine hydrochloride 13*


White solid, 47% yield, *t*_R_ = 4.07, C_24_H_26_Cl_2_N_4_O_2_S, MW 505.46, ^1^H NMR (400 MHz, CD_3_OD) δ (ppm) 1.09 (t, *J* = 7.24 Hz, 3H), 1.69 (s, 3H), 1.82–1.98 (m, 1H), 2.36 (d, *J* = 1.56 Hz, 1H), 2.48 (d, *J* = 1.57 Hz, 1 H), 2.94–3.09 (m, 2H), 3.18 (d, *J* = 2.54 Hz, 1H), 3.46 (dd, *J* = 13.69, 5.48 Hz, 1H), 3.80 (dd, *J* = 13.89, 4.89 Hz, 1H), 7.00–7.11 (m, 2H), 7.21–7.33 (m, 2H), 7.34–7.43 (m, 1H), 7.47–7.56 (m, 2H), 7.64 (s, 1H), 7.76–7.83 (m, 1H), 8.60 (dd, *J* = 8.41, 0.78 Hz, 1H); ^13^C NMR (100 MHz, CD_3_OD) 11.55, 22.29, 28.00, 43.04, 49.18, 53.35, 64.59, 105.95, 114.42, 115.99, 121.63, 122.89, 125.03, 126.27, 128.39, 131.01, 134.28, 135.18, 135.27, 139.54, 146.19, 151.92. Monoisotopic mass 468.14, [M + H]^+^ = 469.4.


*(S)-1-((3-chlorophenyl)sulfonyl)-N-((1-ethylpyrrolidin-2-yl)methyl)-1H-pyrrolo[3,2-c]quinolin-4-amine hydrochloride 14*


White solid, 50% yield, *t*_R_ = 4.08, C_24_H_26_Cl_2_N_4_O_2_S, MW 505.46, ^1^H NMR (500 MHz, CD_3_OD) δ (ppm) 1.26 (dt, *J* = 11.0, 5.5 Hz, 3H), 1.25–1.30 (s, 1H), 1.82–1.95 (s, 3H), 2.00–2.15 (s, 1H), 2.60–2.72 (s, 1H), 3.12–3.32 (m, 2H), 3.33–3.44 (s, 1H), 3.55–3.70 (m, 1H), 3.90 (dd, *J* = 14.0, 5.1 Hz, 1H), 4.53–4.60 (s, 1H), 7.12–7.17 (s, 1H), 7.19 (dd, *J* = 16.8, 9.0 Hz, 1H), 7.40–7.51 (m, 2H), 7.55–7.70 (m, 3H), 7.77–7.80 (s, 1H), 7.95 (d, *J* = 2.6 Hz, 1H), 8.75 (d, *J* = 8.5 Hz, 1H); ^13^C NMR (126 MHz, CD_3_OD) 11.57, 22.31, 28.02, 43.07, 49.22, 53.34, 64.62, 105.96, 114.42, 116.00, 121.66, 122.89, 125.05, 126.29, 128.41, 131.02, 134.30, 135.21, 135.30, 139.56, 146.21, 151.94. Monoisotopic mass 468.14, [M + H]^+^ = 469.4.


*(R)-1-((3-chlorophenyl)sulfonyl)-N-((1-ethylpyrrolidin-3-yl)methyl)-1H-pyrrolo[3,2-c]quinolin-4-amine hydrochloride 15*


White solid, 52% yield, *t*_R_ = 4.09, C_24_H_26_Cl_2_N_4_O_2_S, MW 505.46, ^1^H NMR (500 MHz, CD_3_OD) δ (ppm) 1.22–1.29 (m, 1H), 1.31–1.41 (m, 3H), 1.85–2.12 (m, 1H), 2.46 (dd, *J* = 6.73, 5.30 Hz, 1H), 2.90–3.05 (m, 1H), 3.06–3.19 (m, 1H), 3.63 (dd, *J* = 5.87, 3.87 Hz, 1H), 3.66–3.74 (m, 1H), 3.74–3.83 (m, 1H), 3.83–3.90 (m, 1H), 3.83–3.90 (m, 1H), 3.91–4.04 (m, 1H), 3.92–3.95 (m, 1H), 7.45–7.51 (m, 1H), 7.51–7.57 (m, 2H), 7.62–7.71 (m, 2H), 7.83 (dd, *J* = 8.02, 1.15 Hz, 1H), 7.94 (t, *J* = 2.00 Hz, 1H), 8.17 (d, *J* = 3.72 Hz, 2H), 8.79–8.87 (m, 1H); ^13^C NMR (126 MHz, CD_3_OD) δ (ppm) 40.57, 44.46, 50.21, 56.29, 106.54, 115.09, 123.99, 125.68, 126.92, 130.12, 130.60, 131.56, 135.29, 138.78. Monoisotopic mass 468.14, [M + H]^+^ = 469.4.


*(S)-1-((3-chlorophenyl)sulfonyl)-N-((1-ethylpyrrolidin-3-yl)methyl)-1H-pyrrolo[3,2-c]quinolin-4-amine hydrochloride 16*


White solid, 52% yield, *t*_R_ = 4.10, C_24_H_26_Cl_2_N_4_O_2_S, MW 505.46, ^1^H NMR (500 MHz, CD_3_OD) δ (ppm) 1.22–1.30 (m, 1H), 1.30–1.31 (m, 1H), 1.32–1.41 (m, 3H), 1.34–1.35 (m, 1H), 1.85–2.10 (m, 1H), 2.28–2.52 (m, 1H), 2.52–2.54 (m, 1H), 2.53–2.53 (m, 1H), 2.90–3.05 (m, 1H), 3.05–3.18 (m, 1H), 3.57–3.66 (m, 1H), 3.71 (dd, *J* = 5.30, 0.72 Hz, 1H), 3.75–3.87 (m, 1H), 3.76–3.86 (m, 1H), 3.87–4.02 (m, 2H), 7.46–7.52 (m, 1H), 7.52–7.56 (m, 2H), 7.62–7.71 (m, 2H), 7.80–7.86 (m, 1H), 7.94 (t, *J* = 1.86 Hz, 1H), 8.17 (d, *J* = 3.72 Hz, 2H), 8.10–8.16 (m, 1H), 8.83 (d, *J* = 8.59 Hz, 1H); ^13^C NMR (126 MHz, CD_3_OD) δ (ppm) 40.42, 44.90, 50.19, 56.30, 106.48, 113.13, 115.09, 119.07, 124.00, 126.92, 130.11, 130.60, 131.55, 135.29, 135.76, 138.78. Monoisotopic mass 468.14, [M + H]^+^ = 469.4.


*(R)-1-((3-chlorophenyl)sulfonyl)-N-((1-isobutylpyrrolidin-2-yl)methyl)-1H-pyrrolo[3,2-c]quinolin-4-amine hydrochloride 17*


White solid, 34% yield, *t*_R_ = 4.43, C_26_H_30_Cl_2_N_4_O_2_S, MW 533.51, ^1^H NMR (500 MHz, CD_3_OD) δ (ppm) 0.95–1.07 (m, 6H), 1.23–1.30 (m, 1H), 1.26–1.30 (m, 1H), 1.98–2.12 (m, 2H), 2.12–2.13 (m, 1H), 2.12–2.22 (m, 2H), 2.42 (dd, *J* = 12.74, 6.44 Hz, 1H), 3.03–3.15 (m, 1H), 3.79–3.89 (m, 1H), 4.00–4.11 (m, 1H), 4.12–4.23 (m, 1H), 4.34–4.47 (m, 1H), 7.45–7.57 (m, 2H), 7.61–7.70 (m, 3H), 7.83 (d, *J* = 7.73 Hz, 1H), 7.93 (s, 1H), 8.17 (d, *J* = 3.72 Hz, 2H), 8.86 (d, *J* = 8.02 Hz, 1H); ^13^C NMR (126 MHz, CD_3_OD) δ (ppm) 19.83, 22.03, 25.51, 27.09, 29.43, 42.61, 54.79, 63.49, 106.83, 115.20, 123.94, 125.68, 126.89, 130.62, 131.57, 135.27, 135.72, 138.77. Monoisotopic mass 496.17, [M + H]^+^ = 497.4.


*(S)-1-((3-chlorophenyl)sulfonyl)-N-((1-isobutylpyrrolidin-2-yl)methyl)-1H-pyrrolo[3,2-c]quinolin-4-amine hydrochloride 18*


White solid, 34% yield, *t*_R_ = 4.41, C_26_H_30_Cl_2_N_4_O_2_S, MW 533.51, ^1^H NMR (500 MHz, CD_3_OD) δ (ppm) 0.95–1.06 (m, 6H), 1.24–1.31 (m, 1H), 1.96–2.09 (m, 2H), 1.96–2.10 (m, 1H), 2.08–2.22 (m, 2H), 2.09–2.23 (m, 1–H), 2.38 (dd, *J* = 12.74, 6.44 Hz, 1H), 3.09 (dd, *J* = 12.89, 6.01 Hz, 1H), 3.74–3.86 (m, 1H), 3.91–4.00 (m, 1 H), 4.03–4.14 (m, 1H), 4.21–4.36 (m, 1H), 7.45–7.56 (m, 3H), 7.67 (d, *J* = 7.73 Hz, 2H), 7.83 (d, *J* = 8.02 Hz, 1H), 7.92 (s, 1 H), 8.18 (d, *J* = 3.72 Hz, 2H), 8.88 (d, *J* = 8.31 Hz, 1 H); ^13^C NMR (126 MHz, CD_3_OD) δ (ppm) 19.81, 22.02, 25.54, 27.12, 29.46, 42.63, 54.82, 63.52, 106.81, 115.22, 123.97, 125.69, 126.90, 130.64, 131.59, 135.29, 135.75, 138.80. Monoisotopic mass 496.17, [M + H]^+^ = 497.4.


*(R)-2-(2-(((1-((3-chlorophenyl)sulfonyl)-1H-pyrrolo[3,2-c]quinolin-4 yl)amino)methyl)-pyrrolidin-1-yl)ethan-1-ol hydrochloride 19*


White solid, 60% yield, *t*_R_ = 3.86, C_24_H_26_Cl_2_N_4_O_3_S, MW 521.46, ^1^H NMR (500 MHz, CD_3_OD) δ (ppm) 1.78–1.94 (m, 3H), 1.97–2.15 (s, 1H), 2.60–2.76 (s, 1H), 2.80–2.95 (s, 1H), 3.17–3.30 (m, 1H), 3.35–3.47 (m, 2H), 3.59–3.74 (m, 1H), 3.80–3.86 (m, 2H), 3.87–3.92 (m, 1H), 4.50–4.74 (s, 1H), 7.18 (q, *J* = 6.2 Hz, 2H), 7.41 (t, *J* = 7.9 Hz, 2H), 7.56 (d, *J* = 8.0 Hz, 1H), 7.64 (d, *J* = 7.9 Hz, 1H), 7.72–7.76 (m, 1H), 7.77–7.80 (s, 1H), 7.92 (d, *J* = 3.7 Hz, 1H), 8.70 (d, *J* = 8.4 Hz, 1H); ^13^C NMR (126 MHz, CD_3_OD) δ (ppm) 22.53, 29.01, 43.12, 49.99, 52.28, 54.67, 56.29, 106.88, 114.12, 124.75, 125.87, 127.02, 130.12, 130.76, 131.63, 135.31, 139.79. Monoisotopic mass 484.13, [M + H]^+^ = 485.5.


*(S)-2-(2-(((1-((3-chlorophenyl)sulfonyl)-1H-pyrrolo[3,2-c]quinolin-4 yl)amino)methyl)pyrrolidin-1-yl)ethan-1-ol hydrochloride 20*


White solid, 60% yield, *t*_R_ = 3.83, C_24_H_26_Cl_2_N_4_O_3_S, MW 521.46, ^1^H NMR (500 MHz, CD_3_OD) δ (ppm) 1.77–1.95 (m, 3H), 1.99–2.16 (s, 1H), 2.61–2.75 (s, 1H), 2.81–2.94 (s, 1H), 3.15–3.31 (m, 1H), 3.32–3.45 (m, 2H), 3.59–3.75 (m, 1H), 3.81–3.88 (m, 2H), 3.89–3.91 (m, 1H), 4.51–4.75 (s, 1H), 7.19 (q, *J* = 6.2 Hz, 2H), 7.42 (t, *J* = 7.9 Hz, 2H), 7.55 (d, *J* = 8.0 Hz, 1H), 7.65 (d, *J* = 7.9 Hz, 1H), 7.73–7.77 (m, 1H), 7.79–7.81 (s, 1H), 7.95 (d, *J* = 3.7 Hz, 1H), 8.73 (d, *J* = 8.4 Hz, 1H); ^13^C NMR (126 MHz, CD_3_OD) δ (ppm) 22.51, 28.98, 42.13, 49.89, 52.31, 54.87, 56.53, 106.70, 114.15, 124.83, 125.92, 127.01, 130.31, 130.82, 131.61, 135.77, 139.82. Monoisotopic mass 484.13, [M + H]^+^ = 485.5.


*(R)-1-((3-chlorophenyl)sulfonyl)-N-((1-(3-methoxypropyl)pyrrolidin-2-yl)methyl)-1H-pyrrolo[3,2-c]quinolin-4-amine 21*


White solid, 38% yield, *t*_R_ = 4.25, C_26_H_30_Cl_2_N_4_O_3_S, MW 549.51, ^1^H NMR (500 MHz, CD_3_OD) δ (ppm) 1.80–1.98 (m, 5H), 2.05–2.18 (m, 1H), 2.58–2.73 (s, 1H), 2.73–2.90 (s, 1H), 3.16–3.20 (s, 3H), 3.20–3.27 (m, 1H), 3.33–3.47 (m, 4H), 3.48–3.59 (m, 1H), 3.60–3.79 (m, 1H), 3.91 (dd, *J* = 14.2, 5.5 Hz, 1H), 7.16–7.26 (m, 2H), 7.46 (t, *J* = 8.0 Hz, 2H), 7.56–7.62 (m, 1H), 7.68 (d, *J* = 8.2 Hz, 2H), 7.81 (t, *J* = 1.9 Hz, 1H), 7.97 (d, *J* = 3.7 Hz, 1H), 8.76 (d, *J* = 8.4 Hz, 1H); ^13^C NMR (126 MHz, CD_3_OD) δ (ppm) 22.95, 27.22, 42.79, 53.88, 57.42, 69.30, 101.97, 106.09, 114.61, 115.92, 120.53, 122.26, 123.10, 124.03, 125.22, 126.55, 127.38, 128.72, 129.83, 131.18, 133.80, 134.48, 135.31, 139.54, 152.31. Monoisotopic mass: 512.16, [M + H]^+^ = 513.4.

### 3.2. In Silico Evaluation

#### 3.2.1. Structures of the Receptors

The structure of D_3_R in the complex with antagonist eticlopride (PDB code: 3PBL) and 5-HT_6_R in the complex with agonist serotonin (PDB code: 7XTB) were retrieved from the Protein Data Bank [34].

#### 3.2.2. Molecular Docking

The 3-dimensional structures of the ligands were prepared using LigPrep v3.6 [35], and the appropriate ionization states at pH=7.4 ± 1.0 were assigned using Epik v3.4 [36,37]. The Protein Preparation Wizard was used to assign the bond orders and appropriate amino acid ionization states and to check for steric clashes. The receptor grid was generated (OPLS4 force field) by centering the grid box with a size of 12 Å on the D3.32 side chain. Automated flexible docking was performed using Glide v6.9 [38,39] at the SP level, and ten poses per ligand were generated. All ligands were docked using the induced fit docking (IFD) [40] protocol with SP with an OPLS4 force field [41]. The L-R complexes selected in the IFD procedure were next used in molecular dynamics simulations.

#### 3.2.3. Molecular Dynamics

A 100 ns long molecular dynamics (MD) simulation was performed using Schrödinger Desmond software [42]. Each ligand–receptor complex was immersed into a POPC (309.5 K) membrane bilayer, the position of which was calculated using the PPM web server (https://opm.phar.umich.edu/ppm_server, accessed 20 May 2022) [43]. The system was solvated by water molecules described by the TIP4P potential and the OPLS4 force field was used for all atoms. An amount of 0.15 M NaCl was added to mimic the ionic strength inside the cell. The output trajectories were hierarchically clustered into 10 groups according to the ligand using the trajectory analysis tool from Schrödinger Suite. Based on obtained trajectories, the mean geometrical parameters of the salt bridge (distance and angle) with D3.32 were calculated using the Simulation Event Analysis tool in Maestro Schrödinger Suite.

### 3.3. In Vitro Pharmacological Evaluation

#### 3.3.1. The 5-HT_6_Rs Affinity Evaluation

##### Cell Culture and Preparation of Cell Membranes for Radioligand Binding Assays

HEK293 cells with stable expression of human 5-HT_6_ receptors (prepared with the use of Lipofectamine 2000) were maintained at 37 °C in a humidified atmosphere with 5% CO_2_ and grown in Dulbecco’s modified Eagle medium containing 10% dialyzed fetal bovine serum and 500 μg/mL G418 sulfate. For membrane preparation, cells were sub-cultured in 150 cm^2^ flasks, grown to 90% confluence, washed twice with phosphate buffered saline (PBS), prewarmed to 37 °C, and pelleted by centrifugation (200× *g*) in PBS containing 0.1 mM EDTA and 1 mM dithiothreitol. Prior to membrane preparation, pellets were stored at −80 °C.

##### Radioligand Binding Assays

The cell pellets were thawed and homogenized in 10 volumes of assay buffer using an Ultra Turrax tissue homogenizer(IKA, Warsaw, Poland), centrifuged twice at 35,000× *g* for 15 min at 4 °C, and incubated for 15 min at 37 °C between centrifugation rounds [10]. The composition of the assay buffers was 50 mM Tris HCl, 0.5 mM EDTA, and 4 mM MgCl_2_. The assays were incubated in a total volume of 200 μL in 96-well microtiter plates for 1 h at 37 °C. The process of equilibration was terminated by rapid filtration through Unifilter plates with a 96-well cell harvester, and radioactivity retained on the filters was quantified on a Microbeta plate reader (PerkinElmer, Waltham, MA, USA). For displacement studies, the assay samples contained as radioligands (PerkinElmer, USA) 2 nM [^3^H]-LSD (83.6 Ci/mmol). Nonspecific binding was defined with 10 μM methiothepine. Each compound was tested in triplicate at 7 concentrations (10^−10^ to 10^−4^ M). The inhibition constants (*K*_i_) were calculated from the Cheng−Prusoff equation [30]. Results were expressed as means of at least two independent experiments.

#### 3.3.2. Evaluation of Antagonism at Functional Activity on 5-HT_6_Rs

The functional properties of compounds on 5-HT_6_R were evaluated using its ability to inhibit cAMP production induced by 5-CT (1000 nM), a 5-HT_6_R agonist [10]. The compound was tested in triplicate at 8 concentrations (10^−11^ to 10^−4^ M). The level of adenylyl cyclase activity was measured using frozen recombinant 1321N1 cells expressing the human serotonin 5-HT_6_R (PerkinElmer). Total cAMP was measured using the LANCE cAMP detection kit (PerkinElmer), according to the manufacturer’s directions. For quantification of cAMP levels, cells (5 μL) were incubated with a mixture of compounds (5 μL) for 30 min at room temperature in 384-well white opaque microtiter plates. After incubation, the reaction was stopped, and cells were lysed by the addition of 10 μL of working solution (5 μL of Eu-cAMP and 5 μL of ULight-anti-cAMP). The assay plate was incubated for 1 h at room temperature. Time-resolved fluorescence resonance energy transfer (TR-FRET) was detected by an Infinite M1000 Pro (Tecan, Männedorf, Switzerland) using instrument settings from LANCE cAMP detection kit manual(PerkinElmer, Waltham, MA, USA).

#### 3.3.3. Determination of 5-HT_6_R Constitutive Activity at Gs Signaling

Neuroblastoma cells (NG108-15) were grown in DMEM (Dulbecco’s modified Eagle’s medium) supplemented with 10% heat-inactivated fetal bovine serum, 2% HAT (hypoxanthine/aminopterin/thymidine, Life technologies), glutamine, and antibiotics at 37 °C under 5% of CO_2_. cAMP measurement was performed in cells transiently transfected with a construct expressing the CAMYEL bioluminescence resonance energy transfer (BRET) sensor for cAMP [44] (3 μg DNA/million cells) alone or in combination with a plasmid encoding the human 5-HT_6_R (0.5 μg DNA/million cells). Transfection of the NG108-15 cells was conducted in suspension using Lipofectamine 2000, according to the manufacturer’s protocol. Plasmids and lipofectamine were diluted in Opti-MEM Reduced Serum Media (Gibco) and incubated at room temperature for 20 min before being added to the cells. Transfected cells were subsequently plated in white 96-well plates (Greiner) at a density of 50,000 cells per well. Then, 48 h after transfection, cells were washed with PBS containing calcium and magnesium. A triplicate of well was treated with the tested compound diluted in PBS containing calcium and magnesium at concentrations ranging from 0.1 nM to 10 μM. Intepirdine was used as a control for inverse agonist activity. Coelanterazine H (Molecular Probes) was added in each well at a final concentration of 5 μM and incubated at room temperature for 5 min before measuring BRET in a Mithras LB 940 plate reader (Berthold Technologies, Bad Wildbad, Germany). The decrease in CAMYEL BRET induced by the coexpression of the probe with the 5-HT_6_R as compared to the BRET measured in cells expressing the probe alone was used as an index of the 5-HT_6_R constitutive activity.

## 4. Conclusions

To investigate the impact of structural diversification of the amine fragment of the previously reported compound **PZ-1643**, a dual 5-HT_6_R/D_3_R antagonist, the new series of 1*H*-pyrrolo[3,2-*c*]quinolines modified at position 4 with various pyrrolidine-derived moieties was evaluated for the affinity for 5-HT_6_ and D_3_Rs. The selected compounds displayed a higher affinity and more potent antagonist properties for 5-HT_6_R than the previously reported lead compound; however, their affinity for D_3_R was not improved. As observed in the subsequent molecular dynamics simulations, the structural modifications applied, which were favorable for the interaction with 5-HT_6_R, showed a negative impact on the interactions with D_3_R. These effects result from the differences in the distance and angles formed between the basic center of the molecule and the respective residues of aspartic acid in the receptor binding sites. These changes in the geometry parameters affected the quality of the formed SB. The outcomes of this study provide structural hints for designing of dual-acting 5-HT_6_R/D_3_R antagonists to evaluate a contribution of the combination of 5-HT_6_R antagonism and D_3_R antagonism to the neurodegenerative processes.

## Data Availability

The data presented in this study are available in the Appendix A.

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
