# Peer review of "Impact of the Substitution Pattern at the Basic Center and Geometry of the Amine Fragment on 5-HT_6_ and D_3_R Affinity in the 1*H*-Pyrrolo[3,2-*c*]quinoline Series"

_molecules, 2023, doi:10.3390/molecules28031096_

Round 1

Reviewer 1 Report

Dear authors,

It is with pleasure reading your manuscript. I found your manuscript well written and adequately explains the methods, and the results are well explained. The references are relevant to the theme and relatively new. Synthesis and in silico evaluation are well-designed and explained.

My only concern is the introduction which needs to emphasize the importance of your work. Additionally, proofreading is needed.

Best regards,

Author Response

The reply to the review is uploaded as a Word file.

Reviewer 2 Report

The author has developed a series of compounds to understand the Impact of the substitution pattern at the basic center and geometry of the amine fragment on 5-HT6 and D3R affinity. The author has provided a coherent introduction. A series of compounds were synthesized and evaluated for the binding activity of 5HT6 and D3R. Followed by evaluation for 5HT6 antagonist activity in a cellular assay. The author found that compounds show higher affinity for 5HT6 but not D3R. The author utilized molecular dynamics to understand why compounds show 5HT6 affinity but not D3R. Overall, it looks well-written manuscript focusing on understanding the effect of the geometry of amine fragments on 5HT6 and D3R activity. This manuscript can be accepted after the author addresses following comments.

1)    Why D3R activity was not determined for all compounds? Provide a few lines of explanation on why Ki values were determined for 5HT6 whereas, for D3R percentage binding was determined.

2)    why is control PZ-1643 D3R activity presented as 7nM whereas, other compounds are represented in percentage?

3)    Overall manuscript sections are a bit confusing. Section 1 is an introduction; section 2 is results and section 4 is results and discussion. I would suggest providing suitable titles for each section to make it easy for readers to understand.

Author Response

(The authors gave the same response as above.)

Reviewer 3 Report

In this manuscript, the authors report structure-activity studies on a series of 1H-pyrrolo[3,2-c]quinoline derivatives as potential antagonists of two receptors 5-HT6R and D3R both implicated in neurogenerative and psychiatric disorders.

The rational of the study is well presented in the Introduction. The study is presented as a complement to preliminary former results. Nevertheless many errors appear in the different references given as background bases for the present work. In Reference 19, only indole derivatives are presented with no indication of any pyrroloisoquinoline derivatives.

Another problem occurs with Ref 25 and 26. Compound CPPQ is described in Ref 25 as compound 14 but is only designated as such in Ref 26. Nevertheless, compound CPPQ does not bear any isobutyl chain on the pyrrolidine heterocycle as indicated in the text (line 68). Authors must carefully check their references and properly allocate them in the manuscript. Moreover, CPPQ is used as a reference compound in Figure 3, its exact structure has to be given in the manuscript with the exact reference.

In the Results section, the chemistry part affording compound 5 is an improvement of a first scheme presented in Ref 25 and has already been described in Ref 27. All this preliminary part might be deleted with only reference to Ref 27. To be noted compound 1 is not converted in “respective formaldehyde” but formamides in order to further afford isocyanide 2. Line 95: change access by excess.

In Scheme 2, compounds 15 and 16 only differ by the configuration of one carbon, R might be replaced by ethyl.

The structure activity discussion part is based on binding data and in silico studies. It allows the authors to propose a conclusion which appear to be sound.  

Other comments:

Line 165: The word “Functionnalisation” to describe the change of an ethyl by an isobutyl group does not appear to be adequate and might be changed as well as the word “and” line 172 by “or”.

In the experimental part, the denomination of compound 9 has to be corrected since 8 and 9 are not enantiomers, the carbon on the pyrrolidine heterocycle remains with the S configuration.

Author Response

(The authors gave the same response as above.)
